# Neural Posterior Domain Randomization

**Fabio Muratore**[1,2], **Theo Gruner**[1], **Florian Wiese**[1],
**Boris Belousov**[1], **Michael Gienger**[2], **Jan Peters**[1]

[1] Intelligent Autonomous Systems Group, Technical University Darmstadt, Germany
[2] Honda Research Institute Europe, Offenbach am Main, Germany
Correspondence to `fabio@robot-learning.de`

**Abstract:** Combining domain randomization and reinforcement learning is a widely used approach to obtain control policies that can bridge the gap between simulation and reality. However, existing methods make limiting assumptions on the form of the domain parameter distribution which prevents them from utilizing the full power of domain randomization. Typically, a restricted family of probability distributions (e.g., normal or uniform) is chosen a priori for every parameter. Furthermore, straightforward approaches based on deep learning require differentiable simulators, which are either not available or can only simulate a limited class of systems. Such rigid assumptions diminish the applicability of domain randomization in robotics. Building upon recently proposed neural likelihood-free inference methods, we introduce Neural Posterior Domain Randomization (NPDR), an algorithm that alternates between learning a policy from a randomized simulator and adapting the posterior distribution over the simulator's parameters in a Bayesian fashion. Our approach only requires a parameterized simulator, coarse prior ranges, a policy (optionally with optimization routine), and a small set of real-world observations. Most importantly, the domain parameter distribution is not restricted to a specific family, parameters can be correlated, and the simulator does not have to be differentiable. We show that the presented method is able to efficiently adapt the posterior over the domain parameters to closer match the observed dynamics. Moreover, we demonstrate that NPDR can learn transferable policies using fewer real-world rollouts than comparable algorithms.

**Keywords:** sim-to-real, domain randomization, likelihood-free inference

## 1 Introduction

Learning control policies on a physical robot is time- and resource-intensive. Crucially, Reinforcement Learning (RL) relies on random exploration, which in most cases can not be executed directly on the device. Training in simulation promises to alleviate these problems by generating vast amounts of diverse data faster and cheaper. However, all simulators are only models of reality and therefore guaranteed to be flawed. Thus, using data from a single simulation instance is often not sufficient to learn a control policy which transfers to the real-world counterpart, and might even lead to dangerous overfitting since the optimization process is optimistically biased [1].

The sim-to-real robot learning community suggested numerous approaches to bridge the 'reality gap' within the last few years. Motivated by superior results and flexibility, there is a clear trend towards methods that automatically tune a randomized physics simulator's parameter distribution to closer match the reality [2, 3, 4, 5, 6, 7], also called guided domain randomization. Such approaches generally involve a metric to quantify how well the simulated data is matching the observed data, combined with a mechanism to update the domain parameters or their distribution. Likelihood-Free Inference (LFI) methods measure the closeness between two observations by comparing their (logarithmic) probabilities, and provide various ways to update the density estimator which is either connected to the posterior [8, 9, 10] or the likelihood [11, 12, 13].

5th Conference on Robot Learning (CoRL 2021), London, UK.

**Contributions** We contribute to the state-of-the-art in robot learning by proposing Neural Posterior Domain Randomization (NPDR), an algorithm which intertwines LFI, domain randomization, and RL to infer a distribution over simulators which is subsequently used to train transferable policies. Most notably, NPDR does not make *any* assumptions on the simulator apart from requiring the ability to sample from it, works with coarse priors, and can be configured to strictly yield physically plausible domain parameters. Our approach allows to seamlessly integrate any (parameterizable) physics engine, hence benefits from the rapid progress

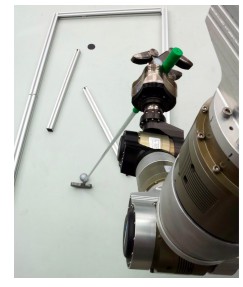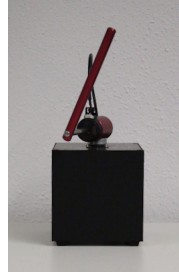

Figure 1: Evaluation tasks: robot mini golf and underactuated swing-up

in the simulation community. Moreover, we synchronize the simulator segmentwise with the trajectories recorded in the target domain. This technique facilitates domain parameter inference for highly dynamical systems. We evaluate NPDR in one sim-to-sim and two sim-to-real experiments, and compare it with (online) BayesSim [14, 15] as well as Bayesian linear regression [16]. Our experiments demonstrate the generality of the proposed approach and show the advantages of flexible posterior inference in challenging continuous control problems. Additionally, we release the source code of our implementations in a open-source library [17].

## 2 Background and Notation

Inferring the dynamics of an environment while solving a task is at the core of model-based RL [18]. We describe this problem using the framework of Markov Decision Processes (MDPs) in which the domain parameters are modeled as random variables with an unknown distribution. This framework is described in Section 2.1. To infer the distribution over the domain parameters without making any additional assumptions on the structure of the model of the MDP, we rely on the LFI methods described in Section 2.2, which only require the ability to sample trajectories from the model.

### 2.1 Markov Decision Processes with Randomized Dynamics

Consider a discrete-time dynamical system

$$\boldsymbol{s}_{t+1} \sim \mathcal{P}_{\boldsymbol{\xi}}(\boldsymbol{s}_{t+1}|\boldsymbol{s}_t, \boldsymbol{a}_t), \quad \boldsymbol{s}_0 \sim \mu_{\boldsymbol{\xi}}(\boldsymbol{s}_0), \quad \boldsymbol{a}_t \sim \pi_{\boldsymbol{\theta}}(\boldsymbol{a}_t|\boldsymbol{s}_t), \quad \boldsymbol{\xi} \sim p(\boldsymbol{\xi}),$$

with the continuous state $\boldsymbol{s}_t \in \mathcal{S}_{\boldsymbol{\xi}} \subseteq \mathbb{R}^{n_s}$ and continuous action $\boldsymbol{a}_t \in \mathcal{A}_{\boldsymbol{\xi}} \subseteq \mathbb{R}^{n_a}$ at time step $t$. The environment, also called domain, is characterized by its parameters $\boldsymbol{\xi} \in \mathbb{R}^{n_\xi}$ (e.g., masses, friction coefficients, or time delays) which are assumed to be random variables distributed according to an unknown probability distribution $p: \mathbb{R}^{n_\xi} \to \mathbb{R}^+$. The domain parameters determine the transition probability density function $\mathcal{P}_{\boldsymbol{\xi}}: \mathcal{S}_{\boldsymbol{\xi}} \times \mathcal{A}_{\boldsymbol{\xi}} \times \mathcal{S}_{\boldsymbol{\xi}} \to \mathbb{R}^+$ that describes the system's stochastic dynamics. The initial state $\boldsymbol{s}_0$ is drawn from the start state distribution $\mu_{\boldsymbol{\xi}}: \mathcal{S}_{\boldsymbol{\xi}} \to \mathbb{R}^+$. Together with the reward function $r_{\boldsymbol{\xi}}: \mathcal{S}_{\boldsymbol{\xi}} \times \mathcal{A}_{\boldsymbol{\xi}} \to \mathbb{R}$, and the temporal discount factor $\gamma \in [0, 1]$, the system forms an MDP described by the tuple $\mathcal{M}_{\boldsymbol{\xi}} = \{\mathcal{S}_{\boldsymbol{\xi}}, \mathcal{A}_{\boldsymbol{\xi}}, \mathcal{P}_{\boldsymbol{\xi}}, \mu_{\boldsymbol{\xi}}, r_{\boldsymbol{\xi}}, \gamma\}$.

The goal of an RL agent is to maximize the expected (discounted) return, a numeric scoring function which measures the policy's performance. The expected discounted return of a policy $\pi_{\boldsymbol{\theta}}(\boldsymbol{a}_t|\boldsymbol{s}_t)$ with the parameters $\boldsymbol{\theta} \in \Theta \subseteq \mathbb{R}^{n_\theta}$ is defined as $J(\boldsymbol{\theta}, \boldsymbol{\xi}) = \mathbb{E}_{\boldsymbol{\tau} \sim p(\boldsymbol{\tau})}\left[\sum_{t=0}^{T-1} \gamma^t r_{\boldsymbol{\xi}}(\boldsymbol{s}_t, \boldsymbol{a}_t)|\boldsymbol{\theta}, \boldsymbol{\xi}\right]$. While learning from experience, the agent adapts its policy parameters. The resulting state-action-reward tuples are collected in trajectories, a.k.a. rollouts, $\boldsymbol{\tau} = \{\boldsymbol{s}_t, \boldsymbol{a}_t, r_t\}_{t=0}^{T-1} \in \mathcal{T}$ with $r_t = r_{\boldsymbol{\xi}}(\boldsymbol{s}_t, \boldsymbol{a}_t)$. When augmenting the RL setting with domain randomization, the goal becomes to maximize the expected (discounted) return for a distribution of domain parameters

$$J(\boldsymbol{\theta}) = \mathbb{E}_{\boldsymbol{\xi} \sim p(\boldsymbol{\xi})}[J(\boldsymbol{\theta}, \boldsymbol{\xi})] = \mathbb{E}_{\boldsymbol{\xi} \sim p(\boldsymbol{\xi})}\left[\mathbb{E}_{\boldsymbol{\tau} \sim p(\boldsymbol{\tau})}\left[\sum_{t=0}^{T-1} \gamma^t r_{\boldsymbol{\xi}}(\boldsymbol{s}_t, \boldsymbol{a}_t)\Big|\boldsymbol{\theta}, \boldsymbol{\xi}\right]\right].$$

The outer expectation with respect to the domain parameter distribution $p(\boldsymbol{\xi})$ is the key difference compared to the standard MDP formulation. It enables the learning of robust policies that work for a whole set of environments instead of overfitting to a particular problem instance.

## 2.2 Sequential Neural Posterior Estimation (SNPE)

A simulator is a generative model $g : \Xi \to \mathcal{X}$ mapping a set of domain parameters $\boldsymbol{\xi} \in \Xi$ to observations $\boldsymbol{x} \in \mathcal{X}$. Generating data from the simulator can be interpreted as sampling from an intractable likelihood $\boldsymbol{x} \sim p(\boldsymbol{x}|\boldsymbol{\xi})$. Given a prior belief $p(\boldsymbol{\xi})$ over the domain parameters and an observation $\boldsymbol{x}^{\mathrm{obs}}$, LFI provides a way to learn an approximation $\hat{p}_{\boldsymbol{\phi}}(\boldsymbol{\xi}|\boldsymbol{x})$ of the underlying true domain parameter distribution $p(\boldsymbol{\xi}|\boldsymbol{x} = \boldsymbol{x}^{\mathrm{obs}})$, where $\boldsymbol{\phi} \in \Phi$ are the parameters of the density estimator, e.g., a neural network or a mixture model. When operating on time series data, LFI requires an embedding to construct features of constant size. This function $f : \mathcal{T} \to \mathcal{X}$ computes features, also called observations, from the rollouts $\boldsymbol{\tau} \in \mathcal{T}$. It may depend on parameters $\boldsymbol{\psi} \in \Psi$ which state-of-the-art implementations optimize jointly with the parameters $\boldsymbol{\phi}$ of the density estimator.

Neural Posterior Domain Randomization (NPDR), as introduced in Section 3, is agnostic to the inference subroutine. In this paper, a neural LFI algorithm SNPE-C [10] is used for its superior performance compared to the alternative approaches described in Section 5.3. SNPE-C approximates the true posterior using normalizing flows, a generative model which produces tractable distributions where both sampling and density evaluation can be efficient and exact [19]. To overcome the intractability of the likelihood, the posterior is trained to minimize the following objective function

$$\mathcal{L}(\boldsymbol{\phi}) = \sum_{r=1}^{R} \sum_{n=1}^{N} -\log \left( \frac{1}{Z(\boldsymbol{x}, \boldsymbol{\phi})} \frac{\tilde{p}(\boldsymbol{\xi}_{r,n})}{p(\boldsymbol{\xi}_{r,n})} q_{\boldsymbol{\phi}}(\boldsymbol{\xi}_{r,n}|\boldsymbol{x}_{r,n}) \right), \quad \boldsymbol{x}_{r,n} \sim p(\boldsymbol{x}|\boldsymbol{\xi}_{r,n}), \quad \boldsymbol{\xi}_{r,n} \sim \tilde{p}(\boldsymbol{\xi})$$

where $R$ is the number of inference rounds $N$ is the number of samples per round, and $Z$ is the normalization constant. Note that the domain parameters are sampled from a proposal prior distribution $\tilde{p}(\boldsymbol{\xi})$, which is initialized with the prior $p(\boldsymbol{\xi})$, and requires importance reweighting of the posterior except for the first round. Sampling from the proposal prior has the advantage that the domain parameters could be sampled in more narrow regions where the actual support of the domain parameter distribution lies. In a multi-round setup, the proposal prior $\tilde{p}(\boldsymbol{\xi})$ is set to be the posterior of the previous round conditioned on the previously observed data $\hat{p}_{\boldsymbol{\phi}}(\boldsymbol{\xi}|\boldsymbol{x} = \boldsymbol{x}^{\mathrm{obs}})$.

## 3 Neural Posterior Domain Randomization (NPDR)

The goal of NPDR is to learn a control policy in simulation such that it transfers to the real device, making as few assumptions as possible about the properties of the simulator, e.g., not assuming differentiability or use of rigid-body dynamics. To achieve this, NPDR augments the nominal RL task with domain randomization and leverages LFI methods powered by normalizing flows to approximate the posterior over the domain parameters. The complete procedure is described in Algorithm 1.

NPDR does not impose *any* restrictions on the inference subroutine (Line 11) or on the policy optimization subroutine (Lines 2 and 13). A key feature of NPDR is the ability to use the current belief over the simulation parameters at every iteration. This belief is inferred such that it explains the data observed from the real system best. Furthermore, the algorithm was implemented in a way that it is possible to jointly condition on multiple real-world rollouts (Lines 6 and 10). To execute the inference (Line 11), we integrated the sbi toolbox [20], which enables us to easily switch between LFI methods. The sbi package expects all observations used to condition the posterior to have the same size. Satisfying this constrain for arbitrary time series embeddings, requires the rollouts to be of equal length. Thus, if a policy causes an emergency stop, e.g., by exceeding the state boundaries, we pad the remainder of the rollout with zeros. During our experiments, we did not observe any negative effect of this measure on the inference routine.

Complementarily, the rollouts are simulated segmentwise, i.e., the states are synchronized with the real trajectory every $K$ time steps. This repeated synchronization is especially beneficial for systems with a fast dynamics like the Furuta pendulum (Figure 1), since the trajectories diverge quickly even for well-fit parameters. Finally, if the policy has an internal model, e.g., true for MPC or the energy-based control law used in Section 4.3, NPDR allows to set the parameters of the internal model to the most likely sample from the current posterior (Line 13). Updating the controller's model was also done in [15], however not in combination with updating the policy parameters.

**Algorithm 1:** Neural Posterior Domain Randomization (NPDR)

---

**input** : initial policy $\pi_{\boldsymbol{\theta}}(\boldsymbol{a}|\boldsymbol{s})$, generative model $g^{\text{sim}}(\boldsymbol{\xi})$, physical device $g^{\text{real}}$, prior $p(\boldsymbol{\xi})$,
   initial density estimator $q_{\boldsymbol{\phi}}(\boldsymbol{\xi}|\boldsymbol{x})$, summary statistics $f_{\boldsymbol{\psi}}(\boldsymbol{\tau})$, LFI subroutine INFER,
   policy optimization subroutine POLOPT,

**output:** approximate (conditional) posterior over the domain parameters $\hat{p}_{\boldsymbol{\phi}}(\boldsymbol{\xi}|\boldsymbol{x}=\boldsymbol{x}^{\text{real}})$,
   policy $\pi_{\boldsymbol{\theta}^{\star}}(\boldsymbol{a}|\boldsymbol{s})$ trained using the approximate (conditional) posterior

1  Initialize the parameters $\boldsymbol{\theta}$, $\boldsymbol{\phi}$, $\boldsymbol{\psi}$ randomly
2  Train an initial policy $\boldsymbol{\theta}^{\star} \leftarrow \text{POLOPT}[\boldsymbol{\theta}, p(\boldsymbol{\xi})]$ with samples from the prior $\boldsymbol{\xi} \sim p(\boldsymbol{\xi})$
3  Initialize the approximate posterior with the prior $\hat{p}_{\boldsymbol{\phi}}(\boldsymbol{\xi}|\boldsymbol{x}) \leftarrow p(\boldsymbol{\xi})$
4  **for** *each iteration* $i = 1:I$ **do**                              ▷ Sim-to-real loop
5  |  Execute $H$ target domain rollouts $\boldsymbol{\tau}^{\text{real}} \leftarrow g^{\text{real}}(\boldsymbol{\theta}^{\star})$
6  |  Compute the observations from the rollouts using learned summ. stats. $\boldsymbol{x}^{\text{real}} \leftarrow f_{\boldsymbol{\psi}}(\boldsymbol{\tau}^{\text{real}})$
7  |  **for** *each round* $r = 1:R$ **do**                    ▷ multi-round LFI, first round is amortized
8  |  |  Sample $N$ domain parameters $\boldsymbol{\xi} \sim \hat{p}_{\boldsymbol{\phi}}(\boldsymbol{\xi}|\boldsymbol{x}=\boldsymbol{x}^{\text{real}})$
9  |  |  Sample $N$ associated rollouts $\boldsymbol{\tau}^{\text{sim}} \leftarrow g^{\text{sim}}(\boldsymbol{\theta}^{\star}, \boldsymbol{\xi})$
10 |  |  Compute the observations from the rollouts using learned summ. stats. $\boldsymbol{x}^{\text{sim}} \leftarrow f_{\boldsymbol{\psi}}(\boldsymbol{\tau}^{\text{sim}})$
11 |  |  Optimize the density estimator and embedding $\boldsymbol{\phi}^{\star}, \boldsymbol{\psi}^{\star} \leftarrow \text{INFER}[\boldsymbol{x}^{\text{real}}, \boldsymbol{x}^{\text{sim}}, \boldsymbol{\phi}, \boldsymbol{\psi}]$
12 |  |  Obtain the cond. posterior from the density estimator $\hat{p}_{\boldsymbol{\phi}}(\boldsymbol{\xi}|\boldsymbol{x}=\boldsymbol{x}^{\text{real}}) \leftarrow q_{\boldsymbol{\phi}}(\boldsymbol{\xi}|\boldsymbol{x}=\boldsymbol{x}^{\text{real}})$
13 |  Train the policy using the latest conditional posterior $\boldsymbol{\theta}^{\star} \leftarrow \text{POLOPT}[\boldsymbol{\theta}^{\star}, \hat{p}_{\boldsymbol{\phi}}(\boldsymbol{\xi}|\boldsymbol{x}=\boldsymbol{x}^{\text{real}})]$

---

## 4  Experiments and Evaluations

We conduct three experiments to analyze the properties of the proposed NPDR algorithm. First, the correctness of the algorithm is evaluated on a simulated environment for which a ground truth solution is available. Namely, a sim-to-sim experiment on the pendulum is conducted to allow for visual inspection of the complete posterior distribution over the domain parameters (Section 4.1). Next, we conduct an experiment in a contact-rich mini golf environment, inferring a 10-dimensional posterior using a state-of-the-art physics engine (Section 4.2). Finally, we integrate policy optimization and solve an underactuated swing-up and balancing task on the Furuta pendulum (Section 4.3). The configurations of all experiments are provided in the appendix.

**Baselines**  Depending on the experiment, we chose BayesSim [14] or its online variant [15] as baseline, because they are the only other approaches that work on the same assumptions and yield a flexible estimator of the domain parameter distribution as well as a trained policy (see Section 5.1 for a detailed description of BayesSim). To obtain comparable results, it was necessary to adapt online BayesSim such that the policy optimization is done until convergence (lines 13–14 of Algorithm 1 in [15]), i.e., we made this part identical to NPDR. For the Furuta pendulum experiment (Section 4.3), we additionally validated the system identification module against Bayesian linear regression.

**Metrics**  In order to assess the quality of the domain parameter posteriors found by the LFI subroutines in NPDR and BayesSim, we compare the state trajectories generated by domain instances sampled from the approximate posterior against the ground truth data that was recorded on the physical system. Before computing the performance metrics, all trajectories are normalized by the state bounds to equalize the importance of the different state dimensions. We decided to use two metrics: the Root-Mean-Square Error (RMSE) and the Dynamic Time Warping (DTW) similarity computed by the python-dtw package [21]. Both metrics are desired to be small. DTW is a dynamic programming algorithm that calculates an optimal match between two temporal sequences [22]. Different DTW methods vary in the cost function and in the assignment of points to each other. We configured DTW to be symmetric, with the beginning and end of the assignment being chosen by the optimizer.

**Code**  The implementations of NPDR as well as the baselines are building on the SimuRLacra framework [17], the sbi toolbox [20], and on the scikit-learn package [23], all open source.

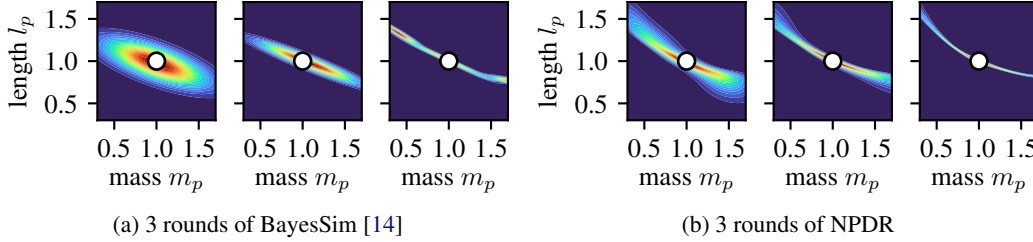

(a) 3 rounds of BayesSim [14]          (b) 3 rounds of NPDR

Figure 2: Learned (full) posterior distributions over the two domain parameters of the pendulum. The LFI rounds are chronologically sorted from left to right. Warmer colors represent higher probabilities. Both algorithms managed to find the manifold of domain parameter configurations which yields observations highly similar ones of the ground truth environment (white circle). Comparing the two sub-figures, we can see benefits of using SNPE-C with a MAF (b) instead of SNPE-A with a MoG (a) to fit the domain parameter posterior.

## 4.1 Sim-to-Sim Validation with a Fully Observable Posterior

The first experiment is set up such that there is a manifold of domain parameters $\Xi^{\text{id}} \subset \Xi$ producing almost identical trajectories $\boldsymbol{\tau}^{\text{id}}$ which are highly similar to the nominal ones. Thus, the observations $\boldsymbol{x}^{\text{id}} = f_{\boldsymbol{\psi}}(\boldsymbol{\tau}^{\text{id}})$ are close to indistinguishable, independent of the summary statistic $f$. The rationale is that the mass $m_p$ and the length $l_p$ always appear coupled in the pendulum's Equations of Motion (EoMs), and are dominated by the term $m_p l_p^2$. Thus, if we only randomize these two domain parameters, then configurations where the product $m_p l_p^2$ has a value similar to the one of the ground truth environment should be inferred as more likely than others. Figure 2 confirms this hypothesis empirically, by clearly capturing the manifold $\Xi^{\text{id}}$. To generate the rollouts, the pendulum has been excited by applying a feed-forward sinusoidal action signal. Both algorithms used the summary statistic function proposed in [14] and the same prior.

## 4.2 Sim-to-Real Bayesian System Identification in a Contact-Rich Environment

One of the main benefits that NPDR and BayesSim inherit from their LFI subroutines is the ability to incorporate state-of-the-art physics engine which expose their parameters. To showcase this advantage, we chose mini golf (Figure 1 and 3) since it requires correct contact modeling to align simulation and reality. We employed the Bullet physics engine to randomize the ball's radius $r_b$, mass $m_b$, restitution coefficient $e_b$, rolling friction coefficient $\mu_b$, the two rails' Cartesian offsets $\Delta x_1, \Delta x_2, \Delta y_1, \Delta y_2$, as well as their angular offsets around the vertical axis $\Delta \gamma_1, \Delta \gamma_2$. The goal location was a circular piece of double-sided tape, modeled as a material with very high rolling

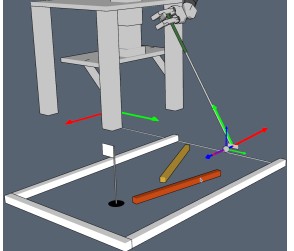

Figure 3: Robot mini golf

friction, that stops the ball quickly. To hit the ball reliably, the robot was driven by a hand-tuned, time-dependent policy. For the inference, we used the ball positions, recorded by a Vicon system, and the robot's joint angles.

Figure 4 displays a slice through the 10-dimensional domain parameter posteriors learned with NPDR and BayesSim on the same data. As expected, the mass of an ideally rolling object has no influence on the dynamics, thus the marginal of $m_b$ should be uniform. The prior for the angular offsets were chosen informatively, i.e., the object can be rotated by $\pm \pi$ at the beginning of the simulation. Therefore, the posterior should have two modes for $\gamma_i$. Both algorithms were able to recognize this multi-modality as well as the coupling between $e_b$ and $r_b$. However, the correlation between $\Delta x_i$ and $\Delta y_i$ was only inferred by NPDR. A quantitative assessment of the modeling accuracy is given in Table 1, which reports the best experiment for every method.

## 4.3 Sim-to-Real Transfer with Policy Optimization in the Loop

The final experiment is aimed at evaluating NPDR on the challenging continuous control task of swinging up and stabilizing a rotary inverted pendulum (Figure 1). This system, known as the Furuta pendulum [24], is particularly hard to control due to its underactuated nature and fast oscillatory

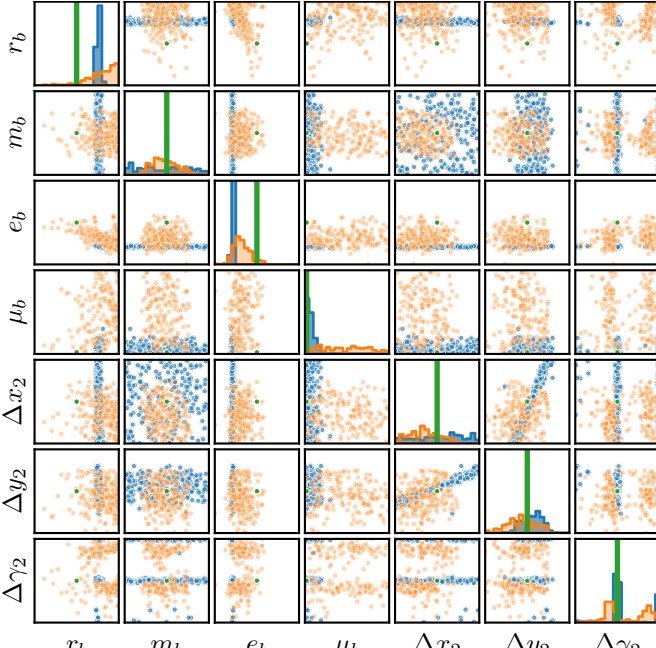

Figure 4: A 7-dimensional slice from the 10-dimensional posteriors learned with NPDR (blue) and BayesSim (orange) in the mini golf experiment. The nominal values (green), were either determined by prior measurements or by coarse estimates. Every domain off-diagonal plot shows the same 200 samples for 2 dimensions, whereas the diagonal plots show the marginal distributions. The omitted domain parameters of the first obstacle $\Delta x_1$, $\Delta y_1$, $\gamma_1$ are similar to $\Delta x_2$, $\Delta y_2$, $\gamma_2$. As described in Section 4.2, the mini golf experiment is set up such that multiple parameter configurations can result in the same trajectory. This arises naturally from broad or non-minimal choices. Details on the domain parameters ranges can be found in the appendix.

movements [25]. To capture a wide range of dynamics, we chose to randomize the link masses $m_r$, $m_p$, the link lengths, $l_r$, $l_p$, the motor constant and resistance $k_m$, $R_m$, the gravity constant $g$, as well as the viscous friction parameters $d_p$, $d_r$ of both joints. As derived in the appendix, it is not possible to linearly separate all of these domain parameters and obtain a unique solution. Therefore, we excluded $g$, $k_m$, and $R_m$ for the Bayesian linear regression, which makes the results baseline comparable to the others. The policy is given as a parameterized hybrid controller consisting of two components: an energy-shaping swing-up controller that brings the pendulum close to the upright position and a stabilizing PD controller that switches on in the vicinity of the upright position [26]. We used Policy learning by Weighting Exploration with the Returns (PoWER) [27] to optimize the policy parameters for all experiments reported in Table 2. During policy execution on the physical device, the parameters of the energy-shaping controller's internal model were set to the most likely domain parameter set, sampled from the posterior during training.

The results show that our LFI-based domain randomization approach outperforms Bayesian linear regression and the nominal baseline, both in terms of the real-world return and the RMSE error. We believe that the advantage of NPDR over BayesSim is due to its superior neural LFI subroutine as well as the fact that our method, in contrast to online BayesSim, trains the policy until convergence before executing the next target domain rollout. This hypothesis is supported by the better metrics for the trajectory differences (Table 2). See Section 5.1 for a more in-depth discussion of the differences between the two algorithms. Comparing the trajectories resulting from the nominal simulation with the ones resulting from the learned distribution (Figure 5), we observe that all approaches improved over the nominal simulator. Moreover, NPDR and BayesSim learned a predictive distribution that includes the ground truth trajectory, except for the segment between $3\,\mathrm{s}$ and $4\,\mathrm{s}$. Since all simulator configurations fail to reproduce the recorded trajectories, we conclude that the modeled EoMs are missing effects which are decisive of the last part of the swing-up.

Table 1: Performances of the Bayesian system identification on the real mini golf environment. The metrics quantify how well each approach fits a common ground truth data set.

| Metric | NPDR | BayesSim | Nominal |
|---|---|---|---|
| DTW dist. | **[3.30 $\pm$ 0.16]e+1** | [9.21 $\pm$ 0.64]e+1 | 1.51e+2 |
| RMSE | **[7.19 $\pm$ 0.65]e−3** | [2.19 $\pm$ 0.51]e−2 | 8.60e−2 |

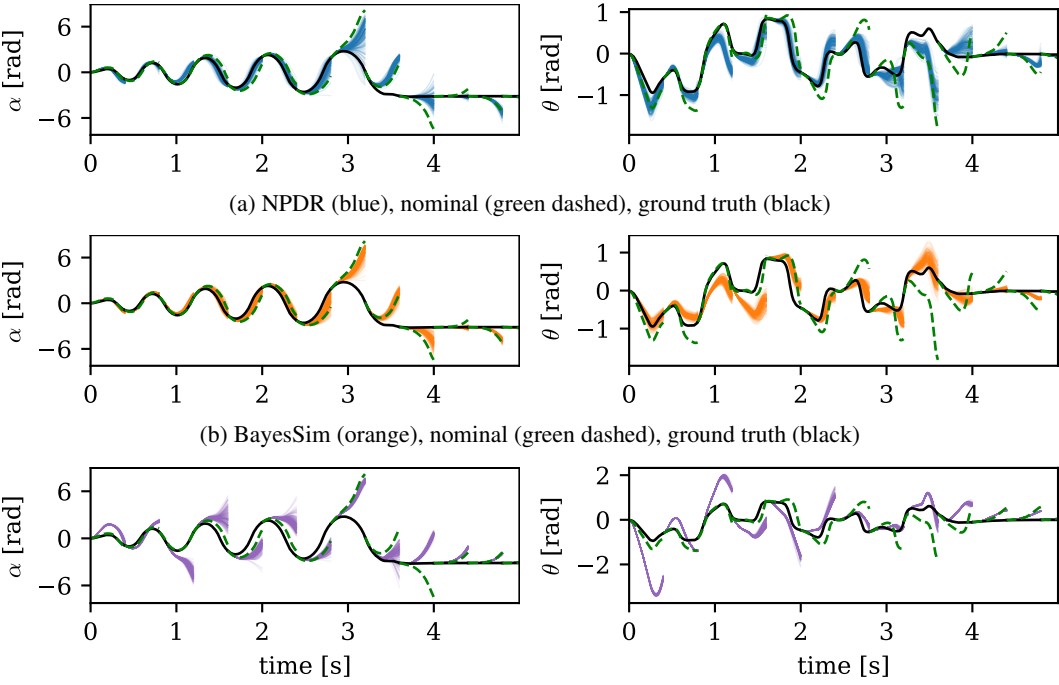

(a) NPDR (blue), nominal (green dashed), ground truth (black)

(b) BayesSim (orange), nominal (green dashed), ground truth (black)

(c) Bayesian linear regression (purple), nominal (green dashed), ground truth (black)

Figure 5: Distributions over trajectories generated from 200 domain parameter sets sampled by the learned posteriors. The simulations were synchronized with the ground truth (black) data every 200 time steps, hence show the feed-forward predictions 200 time steps. The nominal system (dashed green) uses a singe domain parameter set, taken from the manufacturer's data sheet.

## 5 Related Work

The topic of sim-to-real transfer in robot learning has grown in prominence in the last few years. For a broad overview of the existing approaches, we refer the reader to the survey paper [28]. In the following, we focus on the methods directly related to the proposed NPDR algorithm.

### 5.1 Adapting the Source Domain Distribution

The domain parameter distribution can be updated in multiple ways [29, 3, 5, 6, 30]. Most approaches assume independent domain parameters, i.e., they cannot represent the correlations between them. In contrast, NPDR, BayesSim [14], and DISCO [31] model the whole posterior, and thus capture the correlations between the parameters. As observed in the experiments in Section 4, these kinds of correlations are abundant in models of robotic systems. BayesSim [14], especially its online variant [15], share the concepts such as the combination of LFI and domain randomization with NPDR. However, the approaches differ in three important ways: (i) NPDR approximates the posterior using a normalizing flow instead of a mixture model, which is coupled with using SNPE-C instead of SNPE-A as inference algorithm. The results in Section 4 highlight the importance of the

Table 2: Performances of the Bayesian system identification and of the final policies on the real Furuta pendulum. The first two metrics quantify how well each approach fits a common ground truth data set. To quantify the return of a methods, we evaluated 5 (final) policies, generating 10 rollouts per policy. A policy can be considered successful if its return exceeds approximately 1400.

| Metric | NPDR | BayesSim | Bayes. Lin. Reg. | Nominal |
|---|---|---|---|---|
| DTW dist. | **[1.07 ± 0.03]e+3** | [1.24 ± 0.06]e+3 | [2.24 ± 0.04]e+3 | [1.43 ± 0.03]e+3 |
| RMSE | **[2.63 ± 0.04]e−1** | [2.95 ± 0.02]e−1 | [4.46 ± 0.07]e−1 | [3.84 ± 0.03]e−1 |
| Return sim | [9.34 ± 9.35]e+2 | [1.31 ± 3.57]e+2 | **[2.05 ± 0.30]e+3** | **[2.13 ± 0.05]e+3** |
| Return real | **[2.02 ± 0.18]e+3** | [1.11 ± 0.76]e+3 | [3.28 ± 1.44]e+2 | [1.10 ± 0.33]e+2 |

LFI subroutine and clearly favour the neural density estimator. This finding is consistent with [32]. (ii) The policy optimization and domain parameter inference in NPDR are not alternated at every (time) step, but trained until convergence, which stabilizes the training. (iii) We demonstrated that NPDR is able to not only update the policy's internal model, as done with MPC in [15], but also to train the policy parameters directly based on the latest posterior.

## 5.2 Bayesian System Identification for Robotics

A crucial step in NPDR is the approximate inference of the domain parameter distribution. While NPDR makes no assumptions on the simulator properties, other methods provide more domain-specific solutions by assuming more structure in the model. For example, identification of rigid-body dynamics can be framed as a (Bayesian) linear regression problem [33, 34]. However, the identified parameters commonly turn out to be physically implausible [16]. Nonlinear system identification of rigid-body dynamics with neural networks [35, 36] allows for greater flexibility and physical plausibility, but it is not straightforward to obtain confidence estimates based on these methods. For more general dynamical systems, approaches utilizing the classification loss between simulated and real samples have been considered [37, 30]. If the simulator model is non-differentiable, episodic RL has been proposed to optimize the posterior using trajectory similarity as the reward signal [3]. Our NPDR method is the closest to the latter family of algorithms, but instead of episodic RL, it utilizes neural LFI methods, which enable the use of more expressive posteriors, such as normalizing flows, and are more sample efficient, requiring only a few rollouts to be trained.

## 5.3 Likelihood-Free Inference

LFI is a collective term for Bayesian inference methods which are able to learn a posterior distribution in scenarios where the likelihood is either not available or too expensive to evaluate. Approximate Bayesian Computation (ABC) applies Monte-Carlo sampling to infer the parameters by comparing summary statistics of synthetically generated and observed data [38]. Although MCMC-ABC [39] and SMC-ABC [40] enhance the sample efficiency of the classical rejection-ABC by employing more sophisticated sampling schemes, the ABC approach does not scale well because of the inherent problems of sampling in high-dimensional spaces. Amortized methods have been more successful in scaling LFI to more challenging problems by utilizing flexible neural density models [41]. Such models can be used both to generate samples from the posterior and to evaluate the likelihood of parameter configurations. SNPE approaches [8, 9, 10] approximate the conditional posterior, allowing for direct sampling from the posterior, foregoing MCMC. Learning the likelihood [11] can be useful in the context for hypothesis testing. Alternatively, posterior samples can be generated from likelihood-ratios [12] which can be trained using contrastive learning [13]. A tabular summary of state-of-the-art LFI approaches can be found in the appendix. For a comprehensive survey on LFI from simulations, see [42].

## 6 Conclusion

In this paper, we introduced and empirically studied a new adaptive domain randomization algorithm Neural Posterior Domain Randomization (NPDR) that employs normalizing flows for representing the posterior distribution over the domain parameters. Our approach makes no assumptions on the simulator except the ability to sample from it, and it is guaranteed to identify physically plausible parameters. NPDR interleaves approximate posterior inference with policy optimization, leading to an iterative refining of the posterior distribution from which more transferable policies are trained. Our results showed that NPDR improves the flexibility and precision of the existing methods and requires only a few real-world rollouts to train robust policies in a randomized simulation environment. We believe that the proposed method, powered by likelihood-free inference with normalizing flows as density estimators, is a valuable addition to the toolbox of domain randomization algorithms, enabling rapid domain adaptation and training of effective control policies.

Future work will investigate alternative likelihood-free inference approaches, such as sequential neural ratio estimation. Additionally, we plan to apply NPDR to problems involving more challenging dynamical systems such as soft body simulation and fluid dynamics. It is furthermore of interest to consider other modalities of domain parameters, such as parameters of a vision system.

**Acknowledgments**

Fabio Muratore gratefully acknowledges the financial support from Honda Research Institute Europe. Boris Belousov and Jan Peters have received funding from the European Union's Horizon 2020 research and innovation programme under grant agreement No 640554. Calculations for this research were conducted on the Lichtenberg high performance computer of the TU Darmstadt.

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
