# OpenReview forum: "Neural Posterior Domain Randomization"
_robot-learning.org/CoRL/2021/Conference — CoRL2021 Poster_

### Official Review · Reviewer_tnJY · 2021-07-23

**Originality:** Good
**Technical Quality:** Very Good
**Clarity Of Presentation:** Very Good
**Impact:** 3

**Recommendation:**

Weak Accept: I recommend accepting the paper, but will not argue for my recommendation if the majority of other reviewers have a different opinion.

**Summary:**

This paper use SNPE-C (a Likelihood-Free Inference method) to learn a density estimator to compute the posterior conditioned on real observation, and train the RL agent in simulation environment based on the parameters sampled from posterior (domain randomization) to minimize the dynamic sim to real gap, the advantage of the method was shown by one sim2sim and two sim2real experiments.

**Issues:**

1. Clarify the contribution of improvement when apply the SNPE-C to the sim to real compare with the previous paper [1].
2. Show some experiment about how the number of real rollouts influence the performance of the method.
3. It seems the trajectory of each rollout was not provided in paper, it’s hard to evaluate the total number need for real data.


**Reviewer Expertise:**

Good: General knowledge of the area

**Strengths And Weaknesses:**

Strengths:
The method applied in this paper can be applied to simulators that can only sample parameters which is an easy-to-use method. Requirement of real rollout is also reduced by the method, according to experiment setting, only one iteration of training can get good performance, which outperforms than the baseline. Both insight and real experiments are conducted to show the advantage of the method.

Weaknesses:
The method applied in this paper do have a good performance, but this seems mainly because of the strength of the SNPE-C itself, and in a previous paper [1], SNPE-C has already been proposed to compute posterior for simulator. It looks like the improvement for the method is not enough when apply this method to sim2real.

[1] Greenberg, David, Marcel Nonnenmacher, and Jakob Macke. "Automatic posterior transformation for likelihood-free inference." International Conference on Machine Learning. PMLR, 2019.


**Summary Of Recommendation:**

Although the SNPE-C seems just applied from a previous paper, but the previous paper is just use SNPE-C for one simulator, not involve sim2real, and the result in this paper also show good performance. The proposed method has improved a lot compare to the baseline, also considering the low requirement for real data, this maybe a useful method  for dynamic sim2real.

---

> ### Author Response · Authors · 2021-08-25
> **Authors' Answer to Reviewer tnJY**
>
> It is correct that the performance increase comes mostly, but not exclusively, from the usage of SNPE-C with masked autoregressive flows (the policy optimization subroutines, i.e., their hyper-parameters, were chosen to be identical). To investigate this point, we added a small ablation study to Appendix D which shows the influence of the density estimator model. We want to emphasize that for our approach, SNPE-C can easily be replaced with any other inference method that is capable of working with neural density estimators, including methods which are estimating the likelihood, or a likelihood ratio (Table A1 in the appendix). Our choice of SNPE-C was motivated by prior experiments and by the results in [tnJY-1].
>
> In comparison to [tnJY-2], we contribute by demonstrating that SNPE-C can be used to infer the parameters of a state-of-the-art physics simulator based on noisy _real-world_ time series data. Although Greenberg et al. showed in [tnJY-2] that SNPE-C works with time series data, their target domain was given by the Lotka-Volterra model, i.e., a simulation. Furthermore, we believe that it is a contribution to show that SNPE-C is also useful when interleaved with policy optimization for sim-to-_real_ robot learning.
>
> For our experiments, a trajectory contains 800 (simulated pendulum), 800 (robot mini golf), or 1250 (Furuta pendulum) time steps of respectively 2, 16, 4 dimensional states. We found that one trajectory is enough for the LFI procedure to learn a decent posterior estimate. However, the resulting estimate can vary depending on how noisy the measurements are. Thus, more target domain rollouts, e.g., 5 or 10, result in a more accurate posterior which is estimated more consistently over different experiments. On the downside, these additional rollouts come at the price of running the policy on the physical device. In order to visualize the effect of the number of rollouts, we conducted a sim-to-sim experiment on the Furuta pendulum and discussed the results in Appendix D. Finally, we will publish the trajectory data alongside with the code as soon as the double-blind restriction is lifted.
>
> [tnJY-1] J.-M. Lueckmann et al., "Benchmarking Simulation-based inference", AISTATS, 2021.
> [tnJY-2] D.S. Greenberg et al., "Automatic Posterior Transformation for Likelihood-Free Inference", ICML, 2019.

---

### Official Review · Reviewer_FjeG · 2021-07-23

**Originality:** Good
**Technical Quality:** Good
**Clarity Of Presentation:** Very Good
**Impact:** 3

**Recommendation:**

Weak Accept: I recommend accepting the paper, but will not argue for my recommendation if the majority of other reviewers have a different opinion.

**Summary:**

This paper applies neural likelihood-free inference (LFI) to domain randomization, aimed to tackle the sim-to-real transfer problem. LFI brings the benefit of relaxing the restrictive assumptions on the parameter distribution of the simulators and removing the requirement for a differentiable simulator.

**Issues:**

My most major concerns is whether the improvement of the current paper over the method of BayesSim is significant enough, and I have listed the relevant questions to be addressed in Section "Strengths and Weaknesses" above under "Weaknesses" (the first point there). Also, I have some less major concerns/questions which I have also listed under "Weaknesses".

**Reviewer Expertise:**

Good: General knowledge of the area

**Strengths And Weaknesses:**

Strengths:
- The proposed method is a natural step towards more general and better-performing domain randomization for better sim-to-real transfer.
- The paper is well written.

Weaknesses:
- My most major concern is regarding whether the improvement of the current paper over BayesSim is significant enough. Firstly, in lines 230-231, because of this difference (ii), does your proposed method also incur more computational cost? I think an empirical comparison with [14,15] (or a discussion on it if comparison is not feasible) in terms of the computational costs is needed. Secondly, in lines 231-233, why isn't BayesSim [14,15] able to do this? That is, why can't their method also train the policy parameters directly based on the latest posterior? Moreover, I think the performances of BayesSim and the proposed NPDR are actually very similar in Figure 5.
- lines 140-142: I think what is described here are the advantages of the proposed method (i.e., free of potentially restrictive assumptions on the domain parameter distribution, doesn't require a differentiable simulator, etc.), however, it doesn't exclude the comparison with other methods that don't have these advantages (e.g., those methods that assume a parametric distribution on domain parameters). Therefore, I think comparisons with more domain randomization methods should be performed.
- Table 2: why is "Return sim" so high for "Bayes. Lin. Reg."? Do they overfit to the simulations?
- lines 36-39: what's the motivation for using LFI for inferring the domain parameter distribution, instead of other inference methods such as those based on variational inference or MCMC?
- Algorithm 1, line 6 and line 10: I think giving more details on this function $f_{\boldsymbol{\psi}}$ will be helpful

**Summary Of Recommendation:**

This paper improves over the BayesSim method for domain randomization for sim-to-real transfer. My most major concern is the significance of the improvement over BayesSim which determines the technical novelty of the paper, please see details below.

---

> ### Author Response · Authors · 2021-08-25
> **Authors' Answer to Reviewer FjeG**
>
> The vast majority of the computational cost for NPDR as well as BayesSim originates from running the simulations. Once the simulations are done, the posteriors are fitted. Using the sbi toolbox, these weight updates take about 5--300s for SNPE-C as well as SNPE-A. Importantly, SNPE-C is performing a substantially different multi-round inference than SNPE-A, meaning it updates the posterior multiple times on the same set of simulations (unlike SNPE-A). This leads to a notable reduction of the required number of simulations. Eventually, NPDR and BayesSim take approximately the same wall clock time, measured on a desktop PC: 20min (pendulum) 3.5h (mini golf), and 8h (Furuta pendulum). Most of the required time for the experiment on the Furuta comes from the policy optimization and the interaction with the physical device. We added this discussion of the computational cost to Appendix E.
>
> We would like to clarify one misinterpretation: BayesSim can train the policy parameters directly based on the latest posterior. Independent of this, the lines 231--233 (in the first version of our paper) state that online BayesSim does not train the policy until convergence, but updates the domain and policy parameters directly after each other at every time step (lines 13 and 14 in Algorithm 1 in [FjeG-1]). This update strategy did not work at all in our experiments because the random initial policy of online BayesSim did not yield meaningful trajectories for the subsequent inference. We assume that the update scheme of online BayesSim worked in [FjeG-1] for two reasons:
>
> * The planar trajectory tracking task on the wheeled robot in [FjeG-1] is much more benign, especially when in comes to exploration, i.e., even a random policy can gather useful data in this environment.
> * For the experiments in [FjeG-1], BayesSim was not used together with RL, but with MPC (see the last paragraph of Section V. in [FjeG-1]). This means that the control strategy is only updated in terms of its internal model and not by changing the policy parameters.
>
> To obtain comparable results, we adapted online BayesSim to train the policy until convergence the same way we do for NPDR. We noticed that this fact was not explicitly mentioned in the first version of the paper, thus added a corresponding statement in Section 4.
>
> The purpose of Figure 5 is go give the reader a visualization of how samples from the posteriors manifest on trajectory-level, e.g., allowing to gauge when the open-loop predictions start to deviate how much in comparison to the nominal simulation. Figure 5 is not apt to read the performances, since it only displays one posterior per method for one real-world trajectory, both chosen randomly. In order to compare the final policy performances, we refer to the summarized in Table 2.
>
> The paragraph starting at line 140 is intended to explain why we chose these two baselines. In our opinion, real-world evaluations are ultimately the only ones that matter for sim-to-real research. However, evaluating on physical devices is orders of magnitude more expensive in term of time and labor than in simulation. Therefore, we only selected the methods which are the most comparable in the sense that they make similar assumptions. Comparisons against approaches with stronger assumptions on the domain parameter distribution are a definitely good extension for further research.
>
> Bayesian linear regression utilizes the analytic form of the equations of motion and finds a posterior that distributes the probability mass over a wider range of parameter values. We updated Figure A1 in the appendix to illustrate that. In particular, the masses $M_r$ and $M_p$ identified by Bayesian linear regression are significantly different from the values found by NPDR and BayesSim, which makes the simulated environment easier to solve. However, the policy found in such simulated environment does not transfer to the real system as well as the policies found by NPDR and BayesSim.
>
> LFI is motivated by the fact that we do not assume to have access to the likelihood $p(x | \xi)$ (see the lines 87 and 99 in the first version, or the first sentence of Section 5.3). Therefore, variational inference is not feasible. Alternatively, approaches based on ABC and MCMC would also be possible. We decided for a SNPE variant because they had been shown to be superior to 'classical' ABC methods [FjeG-2, FjeG-3, FjeG-4].
>
> We added further information on $f_\psi$ to the lines 6 and 10 of Algorithm 1.
>
> [FjeG-1] R. Possas et al. "Online BayesSim for Combined Simulator Parameter Inference and Policy Improvement", IROS, 2020.
> [FjeG-2] G. Papamakarios and I. Murray, "Fast $\epsilon$-free inference of simulation models with Bayesian conditional density estimation", NIPS, 2016.
> [FjeG-3] F. Ramos et al., "Adaptive domain randomization via probabilistic inference for robotics simulators", RSS, 2019.
> [FjeG-4] J.-M. Lueckmann et al., "Benchmarking Simulation-based inference", AISTATS, 2021.

---

### Official Review · Reviewer_wTxL · 2021-07-26

**Originality:** Fair
**Technical Quality:** Good
**Clarity Of Presentation:** Good
**Impact:** 3

**Recommendation:**

Weak Accept: I recommend accepting the paper, but will not argue for my recommendation if the majority of other reviewers have a different opinion.

**Summary:**

This work proposes a new way to leverage domain randomization to estimate and adapt the posterior distribution over the simulator’s parameters, and consequently allowing optimization of a policy suitable for the target environment in the real world. Specifically, it combines likelihood-free inference methods based on normalizing flows with domain randomization to estimate the posterior. This makes fewer assumptions about the domain parameter distribution, and can model correlations between parameters. This approach is evaluated in a sim-to-real experiment on a swing-up pendulum task.

**Issues:**

- In terms of writing organization, having the related work earlier in the paper such as after the introduction would help a reader who has less familiarity with domain randomization better understand the contributions and differences of this work.

- Describing what the Dynamic Time Warping (DTW) metric is in more detail would make the paper more self-contained.

- As mentioned earlier, it would be useful to include the sample mean and standard deviation of the results across multiple experiments in Table 1, in addition to reporting the max over experiments.


**Reviewer Expertise:**

Fair: Some knowledge of the area

**Strengths And Weaknesses:**

Strengths:
- NPDR combines a strong LFI method with domain randomization to flexibly model the posterior distribution over parameters given a small set of observations. The experiments demonstrate the wide applicability of the method, and also include results on a real pendulum system.

- The paper is overall well written and easy to read. The visualizations help illustrate and characterize the differences between NPDR and BayesSim.

Weaknesses:
- Perhaps, a weakness is that the method is a somewhat straightforward combination of a LFI method based on normalizing flows with domain randomization. However, this work highlights settings that necessitate a stronger neural density estimator.

- It would be useful to include the sample mean and standard deviation of the results across multiple experiments in Table 1, in addition to reporting the max over experiments.

- The studied pendulum control task is quite simple. The returns achieved by NPDR and BayesSim are thus quite close with overlapping intervals. It would be good to apply the method on a wider range of control tasks.

**Summary Of Recommendation:**

I am recommending a weak accept. Overall, the paper clearly communicates the method and its strengths. It also experimentally evaluates several aspects such as its ability to capture correlations between parameters. However, the studied tasks and quantitative results are a bit limited.

---

> ### Author Response · Authors · 2021-08-25
> **Authors' Answer to Reviewer wTxL**
>
> We acknowledge that NPDR is a natural next step for sim-to-real robot learning based on LFI methods. Furthermore, we argue that this paper presents valuable modifications and improvements to the state of the art, thereby reinforcing other Bayesian inference approaches for learning transferable control policies.
>
> We also agree that evaluating over more control tasks is generally good. However, we would like to point out that the swing-up and balancing task on the Furuta pendulum is actually quite challenging because the system is underactuated and the dynamics are fast and oscillatory. Moreover, the phase as well as gain margins for actuation errors are very small. In combination, these properties make it difficult to stabilize at the system's unstable equilibrium point. We also see it as a contribution that we evaluated online BayesSim on a sim-to-real control task that is more challenging than the ones it has been tested on so far (2D trajectory tracking using a wheeled robot). From this perspective, the overlapping intervals verify our implementation of BayesSim, thereby supporting the complete line of research which combines LFI and domain randomization.
>
> To address your second point, we ran 9 additional NPDR and BayesSim experiments for the mini golf task. Each experiment is run with the same hyper-parameters on the same recorded trajectories, but with a different random seed. Based on the new results, we updated Table 1 according to your suggestion.
>
> As requested, we extended the description of DTW to the 'Metrics' paragraph in Section 4.

---

### Official Review · Reviewer_KMoW · 2021-07-27

**Originality:** Fair
**Technical Quality:** Good
**Clarity Of Presentation:** Very Good
**Impact:** 3

**Recommendation:**

Weak Accept: I recommend accepting the paper, but will not argue for my recommendation if the majority of other reviewers have a different opinion.

**Summary:**

This paper presents a method for training policies using Domain Randomization, where the randomization distribution is fit with some real world data. The algorithm, named Neural Posterior Domain Randomization (NPDR) in the paper, consists of using a likelihood-free inference method, Sequential Neural Posterior Estimation with Normalizing Flows (SNPE-C or APT) , to determine the distribution used to train policies in simulation with Domain Randomization. This is similar to BayesSim, which this paper uses as a baseline for comparison.

**Issues:**

The paper compares NPDR to online BayesSim, but not to vanilla BayesSim.

The paper attributes the difference in performance to 1) the choice of inference algorithm 2) the choice of density model 3) that NDPR trains the policy until convergence. These three factors can be removed from NPDR (or added to vanilla BayesSim) independently to determine their importance.

**Reviewer Expertise:**

Excellent: Expert knowledge on the topic of the paper

**Strengths And Weaknesses:**

This paper demonstrates the power of likelihood-free inference methods for the tasks of parameter identification in robotics tasks. Furthermore, it shows how likelihood-free inference can be integrated naturally into a domain randomization pipeline for training robot controllers in simulations. This work also demonstrates the proposed Domain Randomization algorithm on a real physical system, the Furuta pendulum. The paper is clearly written and touches upon a very relevant topic for robotics research. The experimental results on the Furuta Pendulum task show an improvement over the online BayesSim baseline.

The strengths of this paper are in the simplicity of the proposed method and the experimentation with real robot systems.

The weaknesses of this paper are in the originality and impact of its contributions and in the experimental evaluation.

In terms of originality and impact, the proposed algorithm is not very different to BayesSim at a high level. In a detailed view, the proposed algorithm  can be viewed as the same as BayesSim, but replacing the posterior inference algorithm and density model (both of which aren't contributions from this paper).

An important shortcoming of the experimental results is that the differences in experimental results can be attributed to multiple factors (the ones listed in Section 5.2), but the paper does not provide isolated ablations or improvements to the baselines to determine the influence of the components of NPDR in the final performance. The comparison doesn't seem to account for model capacity.

**Summary Of Recommendation:**

While the main technique used in this paper (likelihood free inference with normalizing flows) is of great interest to roboticists, the contribution is thin. I would still recommend this for acceptance given that it would be the first documented application of SNPE with normalizing flows to learning control policies. But I would condition my acceptance to a proper ablation study (to ensure that the method was compared to the best possible baseline)

---

> ### Author Response · Authors · 2021-08-25
> **Authors' Answer to Reviewer KMoW**
>
> The proposed NPDR algorithm is indeed closely related to BayesSim in that it interleaves policy optimization with approximate Bayesian inference of the system dynamics. However, from this perspective, a number of model-based RL algorithms follow such an approach, the difference being in exactly what probabilistic model is used, what inference algorithm, what policy representation and optimization procedure (cf. PILCO, PETS, Planet of the Bayesians, etc.). Therefore, we emphasize precisely the choice of these components and provide evaluations of a novel combination involving normalizing flows as posterior models coupled with SNPE-C as the inference algorithm. The similarity between NPDR and (online) BayesSim is intended and beneficial in order to have a comparable baseline, which allows us to state that our change of the "posterior inference algorithm and density model" is significant.
>
> Moreover, NPDR optimizes the initial policy in simulation using the prior over domain parameters (Line 2 in Algorithm 1). We found that this change was important to achieve the sim-to-real transfer in more challenging control tasks, since the initial policy gathers the first data on the physical system. If the recorded trajectories are informative, the next posterior estimate is very unlikely to be good. Online BayesSim, in contrast, executes a random initial policy (only one gradient step done) on the real robot (Algorithm 1 in [KMoW-1]).
> Another important modification is the segmentwise synchronization of the simulations with the ground truth data, which was necessary since the dynamical system diverge quickly when applying open-loop control (straight-forward application of the recorded actions). We noticed that our description within Section 3 was not placed prominently enough, therefore we added a corresponding statement to the 'Contributions' paragraph.
>
> To the first issue: we did compare NPDR against #vanilla' BayesSim, the pendulum and mini golf experiments (Section 4.1 and 4.2). For these experiments, the rollouts have been collected with a fixed behavioral policy. Therefore, no policy optimization was necessary, i.e., 'vanilla' BayesSim was used.
>
> To the second issue: it is correct that we attribute the difference in performance to these three points, and we added an ablation study to Appendix E which compares the performance of the Bayesian system identification for NPDR (using SNPE-C) with normalizing flows, and a mixture of Gaussians, as well as BayesSim (using SNPE-A) with a mixture of Gaussians. Please note that it is not possible to employ normalizing flows with SNPE-A. The ablation study is evaluating the three alternatives on the same recorded trajectories and does not involve policy optimization. Our results show that using SNPE-C as LFI subroutine yields a better posterior estimate than SNPE-A even when both algorithms use a mixture of Gaussians as the model. Moreover, SNPE-C performs better with a masked autoregressive flow than with a mixture of Gaussians.
>
> [KMoW-1] R. Possas et al. "Online BayesSim for Combined Simulator Parameter Inference and Policy Improvement", IROS, 2020.

---

### Author Response · Authors · 2021-08-25
**Authors' Message to All Reviewers**

First of all, we thank you for investing your time to give us truly helpful feedback. We appreciate the friendly tone in which the reviews have been written, and gave our best to realize the improvement suggestions, address the comments, and answer open questions.

Generally, we want to convey the message that LFI is a powerful tool for sim-to-real robot learning, particularly in combination with domain randomization. While BayesSim is an inspirational approach and performed well on the problems it has been tested on so far, we demonstrate that one can do even better by
* using more efficient inference methods together with more flexible neural models,
* segmentwise synchronization of the simulations (during inference) with the recorded target domain data, and
* training the policies until convergence (crucial for the data collection with the first policy which is trained using the prior).
In this paper, we explicitly connect NPDR (our method) to simulation-based inference approaches, demonstrating the power and flexibility of such methods in the context of robotics and sim-to-real transfer.

Based on your reviews, we did the following changes to the main paper and the supplementary material:
* Updated the contribution statement in Section 1.
* Updated Algorithm 1 to provide more information on $f_\psi$.
* Added a short explanation of the DTW metric in Section 4.
* Added a description of the necessary modification to online BayesSim in Section 4.
* Updated Table 1 with new results, showing the standard deviations for NPDR and BayesSim.
* Fixed a typo in Table 2 where the standard deviation for NPDR was off by one digit. We noticed this when revisiting the data for the ablation study.
* Updated Figure A1 in Appendix D to display the posterior for Bayesian linear regression.
* Added an experiment to Appendix D for NPDR using SNPE-C with a mixture of Gaussians (ablation study).
* Added an experiment to Appendix D for NPDR varying the number of target domain rollouts for domain parameter inference.
* Added a discussion of the computational complexity of NPDR and BayesSim in Appendix E.

---

### Meta-Review · Area_Chair_tU2N · 2021-08-12

**Recommendation:** Accept (Poster)
**Confidence:** 5

**Metareview:**

The paper proposes the use of sequential neural posterior estimation with normalising flows as a likelihood-free inference engine for domain randomisation in robotics tasks. Experiments are provided for both sim-to-sim and sim-to-real with a Furuta Pendulum task.

Reviewers were mostly positive about the paper. The authors are encouraged to reply to all comments from the reviewers, in particular:

1. Better elaborate on the contributions and the differences with respect to the baseline (BayesSim)
2. If possible, provide ablation studies to indicate which factors are affecting performance the most.
3. Short description for DTW and why this metric is more appropriate compared to other divergences such as KL, Wasserstein or MMD.
4. Comparison of computational cost between the proposed and baseline
5. As SNPE is using conditional NFs, the incorporation of the statistical prior leads to a non-analytical posterior (the division of the two priors multiplied by q). How is the new posterior used as a prior in the next iteration?

========= Post discussion update ===========

I thank you the authors for the effort in addressing the reviewers comments and improving the paper. The reviewers agree to accept the paper and this is also my recommendation.

---

> ### Author Response · Authors · 2021-08-25
> **Authors' Answer to Area Chair tU2N**
>
> Thank you for providing a succinct summary of the reviews. Please find the key points of our response below. Apart from combining likelihood-free inference, domain randomisation, and reinforcement learning to close the sim-to-real loop, our paper also demonstrates that the introduced method scales to contact-rich environments as illustrated by the mini golf task.
>
> * We improved the contribution statement in Section 1. Aside from using a more recent LFI method, NPDR differs from BayesSim by synchronizing the simulations during inference with the observed rollouts, and by modifying the policy optimization. Moreover, we actually show the first case where BayesSim is combined with (model-free) reinforcement learning in a closed sim-to-_real_ loop, operating in a higher-dimensional domain parameter space than previous papers using BayesSim.
> * We added an ablation study which compares the performance of the Bayesian system identification for NPDR (using SNPE-C) with normalizing flows, and a mixture of Gaussians, as well as BayesSim (using SNPE-A) with a mixture of Gaussians. The results can be found in to Appendix D.
> * We provided a short description of the DTW metric in Section 4. The DTW metric measures the similarity between two trajectories which may not be perfectly aligned. In contrast, the KL divergence and Wasserstein metric, as well as the MMD are designed to measure the distances between distributions. For a fixed set of simulation parameters, the observed trajectory is a deterministic function of the initial conditions. Therefore, only a metric between two trajectories is needed, and metrics which work on distributions of trajectories are not required.
> * We added a discussion of the computational complexity of NPDR and BayesSim in Appendix E.
> * The purpose of sequentially updating the proposal prior is to find samples, i.e., domain parameters, which are more likely to produce the rollouts collected on the real system. To achieve this, we would like to use the posterior of the previous round (conditioned on the observed data) as our new proposal prior. One benefit of SNPE-C is that the new posterior does not need to be calculated analytically. Instead, SNPE-C approximates the integral in the normalization constant within the training objective by a sum using 'atomic proposals' (Section 3.2 in [tU2N-1]). Thus, we only need to be able to sample from the proposal prior and evaluate the probability of the samples. Since we are employing normalizing flows, both sampling and log-prob evaluation are tractable.
>
>
> [tU2N-1] D.S. Greenberg et al., "Automatic Posterior Transformation for Likelihood-Free Inference", ICML, 2019.

---

### Decision · Program_Chairs · 2021-09-13

**Decision:**

Accept (Poster)

**Comment:**

The paper proposes the use of sequential neural posterior estimation with normalising flows as a likelihood-free inference engine for domain randomisation in robotics tasks. Experiments are provided for both sim-to-sim and sim-to-real with a Furuta Pendulum task.

Reviewers were mostly positive about the paper. The authors are encouraged to reply to all comments from the reviewers, in particular:

1. Better elaborate on the contributions and the differences with respect to the baseline (BayesSim)
2. If possible, provide ablation studies to indicate which factors are affecting performance the most.
3. Short description for DTW and why this metric is more appropriate compared to other divergences such as KL, Wasserstein or MMD.
4. Comparison of computational cost between the proposed and baseline
5. As SNPE is using conditional NFs, the incorporation of the statistical prior leads to a non-analytical posterior (the division of the two priors multiplied by q). How is the new posterior used as a prior in the next iteration?

========= Post discussion update ===========

I thank you the authors for the effort in addressing the reviewers comments and improving the paper. The reviewers agree to accept the paper and this is also my recommendation.